# 'Sometimes you are forced to play God…': a qualitative study of healthcare worker experiences of using continuous positive airway pressure in newborn care in Kenya

Helen M Nabwera [1], Jemma L Wright,[2] Manasi Patil,[1] Fiona Dickinson,[1] Pamela Godia,[3] Judith Maua,[3] Mercy K Sammy,[4] Bridget C Naimoi,[5] Osman H Warfa,[6] Juan Emmanuel Dewez [7] Florence Murila,[8] Alexander Manu,[1] Helen Smith,[9] Matthews Mathai [1]

For numbered affiliations see end of article.

**Correspondence to**
Dr Helen M Nabwera;
Helen.Nabwera@lstmed.ac.uk

## ABSTRACT

**Objective** To explore the experiences of using continuous positive airway pressure (CPAP) in newborn care among healthcare workers in Kenya, and to identify factors that would promote successful scale-up.

**Design and setting** A qualitative study using key informant interviews and focus group discussions, based at secondary and tertiary level hospitals in Kenya.

**Participants** Healthcare workers in the newborn units providing CPAP.

**Primary and secondary outcome measure** Facilitators and barriers of CPAP use in newborn care in Kenya.

**Results** 16 key informant interviews and 15 focus group discussions were conducted across 19 hospitals from September 2017 to February 2018. Main barriers reported were: (1) inadequate infrastructure to support the effective delivery of CPAP, (2) shortage of skilled staff rendering it difficult for the available staff to initiate or monitor infants on CPAP and (3) inadequate knowledge and training of staff that inhibited the safe care of infants on CPAP. Key facilitators reported were positive patient outcomes after CPAP use that increased staff confidence and partnership with caregivers in the management of newborns on CPAP. Healthcare workers in private/mission hospitals had more positive experiences of using CPAP in newborn care as the relevant support and infrastructure were available.

**Conclusion** CPAP use in newborn care is valued by healthcare workers in Kenya. However, we identified key challenges that threaten its safe use and sustainability. Further scale-up of CPAP in newborn care should ensure that staff members have ready access to optimal training on CPAP and that there are enough resources and infrastructure to support its use.

**Ethics** This study was approved through the appropriate ethics committees in Kenya and the UK (see in text) with written informed consent for each participant.

## BACKGROUND

The neonatal period (ie, first 28 days after birth) is well recognised globally as the

### Strengths and limitations of this study

► First study in Kenya that has explored the experiences of healthcare workers using continuous positiveairway pressure (CPAP) in newborn care in-depth, obtaining data from multiple departments with a diverse range of available services and different levels of access to resources.

► The study was timely as it was conducted approximately 2 years after an intense implementation programme of CPAP in Newborn Units in public hospitals, therefore providing key information on the sustainability of this exercise.

► The study was conducted at a time when there were several healthcare workers' strikes that significantly disrupted healthcare services in Kenya, therefore the healthcare worker perspectives in this study may reflect the views of the subset who returned to work after the strikes.

most vulnerable time of childhood.[1] An estimated 2.5 million neonatal deaths occurred worldwide in 2017 accounting for 46% of all under-five deaths.[2] The majority of these neonatal deaths occurred in low- and middle-income countries (LMICs), with the highest burden in South Asia and sub-Saharan Africa (including Kenya).[2 3]

Severe respiratory distress is a serious complication of the three leading causes of neonatal death (ie, preterm birth, intrapartum related events and sepsis) with a case fatality rate of up to 20% in LMICs.[4 5] The provision of respiratory support is therefore a key requirement for the care of hospitalised small and sick newborns, but is often inaccessible in many LMICs.[4 6]

Continuous positive airway pressure (CPAP) is a relatively simple and cheap intervention that saves lives, particularly among preterm or very low birth weight neonates (weight <1500 g) with respiratory distress syndrome.[6–10] CPAP has been used in newborn care in high-income countries (HICs) for over 60 years.[7] The use of CPAP and oxygen therapy in HICs was associated with a 75% reduction in neonatal deaths due to respiratory distress syndrome (RDS).[7] In a trial in Bangladesh, when compared with low-flow oxygen therapy alone, bubble CPAP reduced deaths in children under-five with severe pneumonia and hypoxaemia by 75%.[11] In another trial in Malawi, CPAP reduced deaths by 27% in neonates weighing >1000 g with severe respiratory distress.[12] However, CPAP use is also associated with adverse events, such as nasal trauma, which are rarely reported in LMICs possibly due to staff limitations in being able to identify them.[13–15]

Kenya is one of the 81 countries that account for 90% of child deaths worldwide.[16] Neonatal conditions including severe neonatal infections, birth asphyxia, preterm births and congenital anomalies are the leading cause of under-five mortality.[17] The neonatal mortality rate in Kenya is higher than the overall global estimate (22 per 1000 live births in 2015) and has reduced slowly over the past few decades.[17] To address the deficiencies in neonatal care, CPAP use in newborn units in the public sector has been scaled up since 2014.[18–20] In addition, the Ministry of Health in Kenya recommended the use of CPAP in newborn care in secondary and tertiary level hospitals and developed a national guideline to support its use in 2016.[21] However, factors that influence the sustained and effective use of CPAP in newborn care, relevant to the Kenyan context, are largely unknown.

The aim of our study was therefore to explore the experiences of using CPAP in newborn care among healthcare workers, and to identify the barriers and facilitators of its use, thus informing the future scale-up of this life-saving intervention in Kenya.

## METHODS
Written informed consent was obtained from each participant prior to enrolment.

### Study design
This was a mixed methods study in which we collected both qualitative data and quantitative data from the cross-sectional survey. In this paper, we present the results of the qualitative component of the study in which we conducted key informant interviews and focus group discussions among healthcare workers. This enabled us to gain an in-depth understanding of the gaps in providing CPAP in newborn care and to identify solutions in the context of a devolved, but resource-constrained public healthcare system, with different governance structures.[22–26] The inclusion of private/mission hospitals also

**Table 1** List of newborn baby units with CPAP by Kenyan county

| Name of county | Population of county (millions) | Number of newborn baby units using CPAP |
|---|---|---|
| Nairobi | 3.1 | 6 |
| Kiambu | 1.6 | 2 |
| Machakos | 1.1 | 1 |
| Nakuru | 1.6 | 2 |
| Mombasa | 0.9 | 1 |
| Uasin Gishu | 0.9 | 1 |
| Bomet | 0.7 | 1 |
| Kisumu | 0.3 | 2 |
| Homa Bay | 0.9 | 1 |
| Siaya | 0.8 | 1 |
| Migori | 0.9 | 1 |
| Kisii | 0.2 | 1 |
| Bungoma | 1.4 | 1 |
| Kakamega | 1.7 | 1 |
| Busia | 0.7 | 1 |
| Total | | 23 |

CPAP, continuous positive airway pressure.

enabled us to gain diverse perspectives on CPAP use in Kenya in the context of differing levels of resources.

### Study population and setting
The study population included all healthcare workers involved in providing inpatient secondary or tertiary level newborn care services in Kenyan hospitals. Identification of these hospitals was based on information provided by the Division of Child and Adolescent Health at the Ministry of Health of Kenya and non-governmental organisations (NGOs) involved in newborn care in Kenya. Twenty-three hospitals provided CPAP in newborn care (18 public, 3 private 'for-profit' and 2 mission 'not-for-profit'). The study was conducted across 19 hospitals that were spread across 15 of the 47 counties of Kenya. These included 15 public referral hospitals (8 County, 6 Regional and 1 National level) and 2 private 'for-profit' and 2 mission 'not-for-profit' hospitals. These constituted 83% of all hospitals in Kenya that provided CPAP newborn care. (table 1)

### Sampling
There were two phases of sampling in this study. Convenience sampling was used to recruit the 19 hospitals, which were the hospitals that gave us permission to access their facilities and staff to conduct our study.[27] Purposive sampling was used to recruit healthcare workers with any experience of working on the newborn units that offered CPAP. This enabled us to get a broad range of views and experiences on CPAP use within and between different newborn units in Kenya.[28] We estimated that at each hospital, a sample size of 10 to 12 participants would

**Table 2** Sampling framework

| Number of hospitals using CPAP in newborn care | 23 | | |
|---|---|---|---|
| Type of hospital | 18 public | 3 private | 2 mission |
| Hospitals in study | 15 public | 2 private | 2 mission |

CPAP, continuous positive airway pressure.

be sufficient to achieve data saturation, where no new themes would emerge from collecting more data from the site.[29] We included all hospitals that provided CPAP in their newborn units and were willing to participate in the study. Our sampling framework is outlined in table 2.

The focus group discussions consisted of six to seven healthcare workers with different levels of expertise in newborn care (including nursing students, nurses, medical officers, clinical officers and senior paediatricians), in order to get a perspective of the teamwork and decision-making process around CPAP use. For the key informant interviews, we interviewed senior nurses or doctors who we expected to have an appropriate level of knowledge of the operational aspects of CPAP use in their respective newborn units. As we conducted this work at a time when there were multiple healthcare worker strikes in Kenya, many newborn units were already short staffed. However, we worked closely with the Ministry of Health, county governments and hospitals to ensure that we only collected our data on days when the units reported having adequate numbers of staff and could function for 1 to 2 hours with fewer staff in order to attend our interviews. On many occasions we had to cancel planned visits and wait for a more appropriate time.

### Data collection
Data was collected from September 2017 to February 2018. The key informant interview and focus group discussion guides focussed on the following areas in the context of CPAP use in newborn care: training, decision-making for initiating or stopping, resources and interpersonal relationships. The guides were pilot tested with nurses at one of the county referral hospitals, and this process and data were used to refine the guides. Informed consent was sought from each participant prior to conducting the focus group discussions and key informant interviews. All interviews/discussions were conducted face-to-face at the respective hospitals by six research assistants (five women and one man). All research assistants were current or former practicing clinicians, all of whom had either used CPAP or were trained in its use. All research assistants had received 3 days of intensive training on conducting qualitative interviews/focus group discussions and were fluent in both English and Kiswahili. All interviews were predominately conducted in English, although some participants used terms in Kiswahili to explain their points. The interviews/focus group discussions were recorded and transcribed verbatim by two of the research assistants and an independent transcriber. Each of the research assistants

reviewed the transcripts for accuracy and discussed these with the lead investigator during debriefing sessions to establish the validity of the transcripts.[30]

### Data analysis
We used thematic analysis.[26 31 32] Data analysis was an iterative process that started alongside data collection, allowing exploration of key emerging themes.[32] A conceptual framework was used to guide our analysis (figure 1). After extensive familiarisation with the data, five of the investigators (HMN, MP, JLW, MKS and BCN) developed a coding scheme, which was revised further during the analysis to ensure the robustness of the approach.[33 34] NVivo 11 software (QSR International Pty Ltd 2015) was used for data management.

### Patient and public involvement statement
This was a countrywide qualitative study that was conducted among healthcare workers working on newborn units in hospitals in Kenya in 2017/2018. This study was designed to explore healthcare workers experiences of using CPAP in newborn care in Kenya. No patients or members of the public were involved in the design, recruitment or conduct of the study. The results were disseminated to participants through the leadership in their departments and counties, after the study was completed.

### RESULTS
We conducted 16 key informant interviews and 15 focus group discussions among 100 healthcare workers in Kenya. The majority were nurses (56%) in the age range 31 to 40 years, and 77% were women. The overall median number of years of working in healthcare was 15 years (IQR 8 to 17) and in newborn care was 4 years (IQR 3 to 5) (table 3).

Although healthcare workers felt that the use of CPAP had the potential to enhance newborn care in Kenya, there were multiple challenges with the implementation predominantly in the public sector. The key barriers were provision of inadequate infrastructure for effective delivery of CPAP; shortages of skilled staff that hindered CPAP initiation and management; and inadequate knowledge and training of CPAP among staff limiting its safe use. However, these barriers were sometimes mitigated by positive patient outcomes that increased staff confidence in CPAP use and promoted partnership between caregivers (parents or other key family members) and healthcare workers when initiating CPAP. Healthcare workers in better-resourced private/mission hospitals had more positive experiences of using CPAP in newborn care.

### Effect of inadequate infrastructure on optimal delivery of CPAP
Overall, the infrastructure to support CPAP use in newborn care within the public sector was reported as inadequate. Healthcare workers highlighted that there were often not enough CPAP machines to meet patient

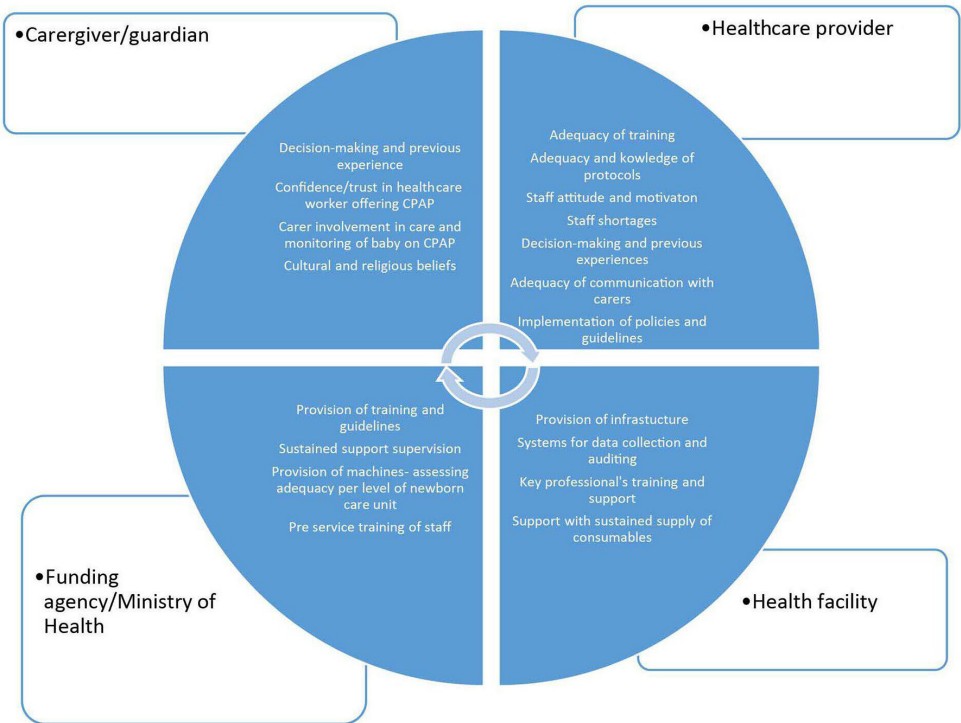

**Figure 1** Conceptual framework used to guide data analysis. CPAP, continuous positive airway pressure.

demands, meaning that they frequently faced the difficult decision of rationing CPAP.

Sometimes you are forced to play God, which we don't like. We would want to have as many CPAP machines as possible but, yeah, it's not possible. So, this is the matter of weighing and looking at who is most likely to survive. (key informant interview, newborn unit 5, paediatrician, public hospital)

Staff described the ethical dilemma of discontinuing CPAP support inappropriately early in a baby that was improving in order to initiate it on a sicker baby. This was extremely distressing for staff.

It is a psychological torture to us and even to them, to the mother. So, such experiences we have had them severally and even for us we feel guilty 'If I could have done this for the baby they could have survived as

| Table 3 | Characteristics of study participants | | | | | |
|---|---|---|---|---|---|---|
| | National referral hospital (Level 6) n=8 | Regional referral hospital (Level 5) n=36 | County referral hospital (Level 4) n=36 | Private n=8 | Mission n=12 | Overall n=100 |
| Majority age group, years | 31–40, 41–50 | 18–30 | 31–40 | 31–40 | 18–30 | 31–40 |
| Women, n (%) | 7 (88) | 26 (72) | 27 (77) | 8 (100) | 9 (75) | 77 (77) |
| Staff cadre | | | | | | |
| Medical students, n | 1 | | | | | 1 |
| Nurses/midwives, n | 5 | 20 | 19 | 6 | 6 | 56 |
| Clinical officers, n | | 4 | 1 | | 1 | 6 |
| Medical officers, n | 3 | 9 | 11 | 1 | 3 | 27 |
| Paediatricians, n | | 3 | 4 | 1 | 2 | 10 |
| Years of experience in healthcare, median (IQR) | 17 (9 to 23) | 15 (4 to 23) | 8 (2 to 20) | 17 (12 to 19) | 2 (2 to 6) | 15 (8 to 17) |
| Years of experience in newborn care, median (IQR) | 4 (3 to 10) | 5 (2 to 10) | 3 (1 to 6) | 5 (4 to 7) | 2 (1 to 4) | 4 (3 to 5) |

these ones', you see. (focus group discussion, newborn unit 8, nurse, public hospital)

There were also frequent shortages of key CPAP consumables in newborn units, such as appropriately sized nasal prongs, due to difficulties with procurement. Healthcare workers reported that after the completion of donor-supported implementation, hospitals struggled to find affordable suppliers for these consumables. The newborn units therefore either improvised or reused equipment that was supposed to be disposed of, posing an infection risk to the newborn and increasing the risk of other adverse events.

Like the nasal prongs size zero is usually a challenge…sometimes they go out of stock…we usually wash them, they are actually supposed to be disposed but we are allowed to wash them even if they are not visibly dirty, but even if you washed them, it reaches that time you must throw them. (focus group discussion, newborn unit 8, nurse, public hospital)

General infrastructure difficulties also meant that use of CPAP was restricted by other issues, such as unreliable power and oxygen supplies, and lack of physical space in the newborn units for the CPAP machinery.

At times our electricity is on and off, I remember a time a baby was on CPAP and our generator had a problem and we were told to wait for around 35 minutes for them to fuel the generator. So, it interfered with that process. (focus group discussion, newborn unit 14, nurse, public hospital)

Another factor is the availability of space, like in our nursery when you come when we have babies, we just put babies close, …and CPAP needs a spacious place. So sometimes you can have a challenge starting it, because where will you put these babies so that you make a space for that child. (focus group discussion, newborn unit 8, nurse, public hospital)

The data also highlighted a lack of other relevant resources required to treat neonates, specifically neonatal medications and laboratory support.

Again, the other challenge is the laboratory investigations, the baby is on CPAP, maybe you will need investigation like blood cultures, but you find that the hospital does not have enough reagents… So, this is a baby on CPAP but you cannot even investigate. (focus group discussion, newborn unit 9, nurse, public hospital)

### Shortages of skilled staff that hindered CPAP initiation and management

Staffing issues across most newborn units were identified as a key barrier to the use of CPAP, either due to staff shortages, a high staff turnover or healthcare worker strikes. It was highlighted that babies on CPAP need extra monitoring by the nursing staff, so staff shortages

were felt to directly limit CPAP use on the newborn units. Experiences included times on busy shifts when the staff did not put babies on CPAP, using oxygen instead due to the extra monitoring required for babies on CPAP, even though they knew this would result in suboptimal care for the baby.

You find you are only two with more than 20 babies in the NBU, you feel if I fix (CPAP), I am still doing other things and I will not be able to observe the baby as required. (focus group discussion, newborn unit 6, nurse, public hospital)

Unfortunately, there were several prolonged healthcare providers' strikes in the public sector during our data collection, which disrupted CPAP use. These strikes resulted in reduced confidence in using CPAP among healthcare providers, many of who had received in-house training but were dependent on resources or senior support lost during the strikes.

We have the cards (containing instructions for setting up CPAP)…With the strike right now I don't know exactly where they are but I think we are going to find them somehow. (focus group discussion, newborn unit 15, nurse, public hospital)

### Inadequate knowledge and training for appropriate CPAP use

The introduction of CPAP to newborn units in the public sector often occurred with support of external NGOs. The acquisition of CPAP machines was accompanied by initial training of the senior members of staff in each unit, with a plan to cascade this training to other staff members during in-house educational meetings.

I think one of the things that has made it work in this hospital is the good training that has been passed on from the pioneer group that was trained. (focus group discussion, newborn unit 17, medical officer, public hospital)

However inadequate training of staff was reported in most of the newborn units as the plan to cascade the training often did not occur. In some cases, the most senior members of the team with skills for using CPAP left and were replaced with new staff members untrained in the use of CPAP. This resulted in a decline in the skills and confidence of using CPAP among healthcare workers in these units to the point that CPAP was no longer used even though the equipment was available and in working order.

Just after the training, I think our paediatrician left the facility and from that time, utilisation and prescription of CPAP was very, very minimal and kept deteriorating until now it is like nobody is thinking about CPAP. (focus group discussion, newborn unit 6, senior nurse, public hospital)

On-going training on CPAP was variable with staff from private/mission hospitals describing regular and

structured training, while most public sector healthcare workers only received informal on-the-job training. This training was very dependent on the level of support for CPAP within the unit and the presence of key educators. From our data we found that training occurred more regularly in private and mission hospitals, where there was access to better resources and fewer admissions.

> Once they get the training, they are very positive and also willing to train others. So, it does not matter whoever is training, so long as you have been trained by somebody else it is something you pass to the next person. (focus group discussion, newborn unit 17, senior nurse, public hospital)

However, even with regular training, the frequent rotation of staff between hospital departments was also noted to have a significant impact on the competencies of the staff.

> One of the biggest challenge is the structure of the healthcare system because the people we started with, the ones who were trained were changed, and a new team comes which is totally not competent and you tell them about CPAP and they fear. (key informant interview, newborn unit 9, public hospital)

Inadequate numbers of staff with skills for using CPAP also hindered staff from initiating this therapy, and healthcare workers reported that many of them had not received training on using CPAP. As a result, some reported witnessing inappropriate cessation of CPAP when a patient was handed over to another member of staff.

> So, if I'm working in the morning, I would be able to use CPAP. There is the other person who comes by night. If they feel they are not comfortable in the use of CPAP, they wouldn't put children on CPAP. (key informant interview, newborn unit 5, public hospital)

The healthcare workers also had limited knowledge about the identification and management of common CPAP complications, resulting in potentially unsafe use of CPAP.

> I will not deny because I have not been taught how to identify complications on CPAP. (focus group discussion, newborn unit 14, medical officer, public hospital)

Notably, the presence of a 'CPAP champion' in a newborn unit was noted to improve the frequency of training and level of knowledge about CPAP among the staff members. Good leadership was critical in ensuring that all staff members were trained to use CPAP.

> We have two champion nurses, we have the nursing officer in charge…she got to understand how they connect, and what are the essential things with the CPAP, then now what she does is she trains any new

nurses who come through. (key informant interview, newborn unit 12, public hospital)

### Positive infant outcomes that increased staff confidence in CPAP use

Staff in newborn units felt that CPAP was useful in improving neonatal care when used appropriately. Indeed, personal experience of good outcomes associated with CPAP use was identified as improving staff morale and their confidence in using CPAP.

> There are some babies we had managed there who had (a) bad prognosis and sometimes you put (them) on CPAP and you see the baby is not in (a) very good state. As I have said the prognosis is bad, but you put on CPAP and you monitor, and they pull through. Even other healthcare providers, they get surprised. (focus group discussion, newborn unit 14, medical officer, public hospital)

However, there was recognition that when it was used inappropriately, for example, when the recommended criteria for initiating or stopping CPAP were not adhered to, the outcomes were poor. This resulted in diminished confidence of CPAP use among staff and had a negative impact on its sustainability in those units.

> It has instilled fear among the nurses such that you take them through the training but when you tell them to go and start, you find that they fear because they don't want to be associated with starting a CPAP and a baby dying, that is now the problem. (focus group discussion, newborn unit 9, nurse, public hospital)

High staff confidence and engagement was identified as important in promoting the use of CPAP. The empowerment of all members of the clinical team to initiate CPAP on newborns who meet the criteria was important in promoting its use, particularly in reducing delays in starting this intervention. This was especially noted if staff members were well supported by the leadership of the hospital.

> The CPAP is good because the nurse can assess, prescribe and administer even without a doctor and when the outcome is so good, it's motivating to the nurses. When you work like that you get so excited you feel you're empowered, and you feel like doing it more and more times. (focus group discussion, newborn unit 8, nurse, public hospital)

### Effect of partnerships between caregivers and healthcare workers on uptake of CPAP

Use of appropriate communication skills that involved the caregivers in the decision-making of care strategies for their infants was felt to enhance acceptability of CPAP.

> Our policy is that they really have to consent, and not just consent in feeling that you are coercing them… Our target is to end up with a win-win, have a mutual

agreement. It makes our work easier, and makes the baby get well faster. Because we are working as a team. (focus group discussion, newborn unit 1, nurse, private hospital)

Caregivers were also reported to be more receptive to the use of CPAP if they were actively encouraged to be involved in the care of their baby while on CPAP.

The mother or the caregiver must be aware, you must counsel them, they accept before you start CPAP because if they refuse then it means they'll interfere with the technical bit of it. (focus group discussion, newborn unit 8, nurse, public hospital)

Overall, in units where the outcomes of the babies on CPAP were good, the healthcare workers reported that both the main caregivers and the wider community became more receptive to CPAP.

It's a bit tricky for…if you put CPAP and a child passes on, nobody will now accept CPAP, no one will accept. If the one improves, the next one also wants it. (focus group discussion, newborn unit 5, nurse, public hospital)

## DISCUSSION

Our study describes experiences of healthcare workers using CPAP within newborn units in Kenya. The implementation gaps were predominantly in the public sector and these were mainly at the health system level. However, we also found factors at a personal level for both caregivers and healthcare providers that facilitated CPAP use, even in the context of resource limitations.

Our study found that CPAP was felt to be beneficial when the right training and infrastructure was in place for its use. This is consistent with findings from an observational study in a rural Ugandan neonatal intensive care unit (NICU), which found that training using the Silverman-Anderson respiratory severity score made it feasible to implement CPAP in newborn care safely in a resource-limited setting.[35] Another study in a rural mission hospital in Kenya showed higher survival-to-discharge rates in premature infants with severe respiratory distress syndrome after the introduction of CPAP with adequate staff training and infrastructure to support its use.[19] In Malawi, a non-randomised controlled study also showed a 27% absolute improvement in survival of neonates with severe RDS[12] and that CPAP was a cost-effective strategy for the care of newborns with respiratory difficulty when compared with nasal oxygen therapy.[36] Our data suggests that poor infrastructure is a major hindrance to the safe and effective use of CPAP in Kenyan public hospitals, especially once external funding to support CPAP implementation including provision of adequate equipment, staff training and tools to monitor it safe use, ends.

The infrastructure and supplies for CPAP use were inadequate particularly in the public sector where

sustaining CPAP use after the completion of donor funded programmes was hampered by the challenges with procurement of consumables. As a result, newborns were potentially being exposed to risks such as infections (due to prolonged use or recycling of respiratory circuits), hence limiting its benefit to these vulnerable babies. In addition, staff had to frequently ration CPAP, which sometimes meant deviation from the national guidelines for stopping CPAP in order to try and meet the demand. In a related study in India, the frustration of the lack of resources to deliver CPAP optimally in newborn care units left staff feeling despondent and demotivated, which had implications for the scale up of this intervention.[37] Indeed, our data from Kenya showed that when staff were not able to deliver CPAP optimally and had newborn mortalities when implementing it, this resulted in fear and anxiety of using CPAP. This fear of CPAP ultimately resulted in decreased use therefore hindering the global strategies to end preventable deaths in newborns.[38]

The inadequate number of skilled staff, particularly nursing staff, was a major challenge. Nurses within our study reported being reluctant to start CPAP use due to poor nurse:patient ratios leading to the inability to monitor these babies adequately while on CPAP—a WHO prerequisite for using CPAP.[8 39] Inadequate staffing of newborn units is a major challenge in Kenya, with the current median nurse:baby ratio in the public sector reported to be as high as 17 babies per nurse.[40] Recent exploratory work in hospitals in Nairobi, Kenya found that nurses had developed strategies to cope with their workload by task-shifting, whereby less technical tasks are shifted to student nurses, other support staff or mothers to free-up nursing time to focus on the more technical aspects of newborn care.[41] Future projects are planned that will seek to formalise this task-shifting to support nurses working in these busy newborn units, which may also have an impact on the delivery of CPAP.[41]

Inadequate training of healthcare workers was another common problem identified within our data that had a negative impact on CPAP use. A recent cross-sectional study from Kenya also found that over one-third of sick newborns were cared for in an environment where nursing knowledge was very low.[42] When CPAP was introduced in public hospitals in western Kenya, a successful training-of-trainers model was used,[18] but our findings suggest that its effectiveness was not sustained over time. Regular staff rotations and prolonged healthcare worker strikes in the public sector appears to have had a negative impact on CPAP-related competencies of staff. In the context of limited training opportunities, work has been done in Malawi to develop a simple clinical algorithm, to identify newborns who would benefit most from CPAP, with evidence of high inter-rater reliability, sensitivity and specificity for their new 'TRY' tool.[43 44] In addition, new training-of-trainers programmes[45] and health partnership models, where international experts support training strategies over a longer period of time,[46] could be beneficial within

Kenya. Sustained refresher training on the use of CPAP in newborn care units in LMICs is also crucial. Indeed, a recent retrospective cohort study in rural Rwanda that found that even after training staff, identification of eligible newborns was a challenge.[47]

Finally, our findings also suggest that partnership between healthcare workers and the main caregivers in the decision-making surrounding CPAP use in their babies is critical to the acceptability of this intervention. Internationally, there is a growing recognition of the importance of involving parents in the care of their hospitalised newborns in improving outcomes for both the infant and their family.[48] A descriptive study conducted at a tertiary newborn unit in Malawi found that caregivers felt anxious and fearful when their infants were on CPAP with limited parent-child interaction.[49] However, this anxiety was mitigated by support from healthcare providers, family and friends.[49] In Rwanda a 'parental neonatal curriculum' has recently been developed using a Delphi consensus technique involving parents and healthcare workers.[50] The development of such curricula has the potential to enhance true partnership with parents.[50] In Malawi, Family-Led Care Models have recently been implemented to improve the facility and home-based care of preterm/low birth weight newborns.[51] The evaluation of the impact of these strategies is keenly awaited.

The key limitation of this study was that it was conducted at a time when there were several healthcare workers strikes that significantly disrupted healthcare services in Kenya. Our findings may therefore reflect the views predominantly of healthcare workers who returned to work after the strikes. As a result, our data predominantly contains the perspectives of nurses, who were the frontline staff involved in delivering CPAP in newborn care in Kenyan hospitals. Unfortunately, we were unable to compare the differing perspectives of doctors and nurses in order to highlight any differences in staff experience in line with their professional background. It would have been useful to interview doctors and nurses independently in order to identifying any potential team working challenges that might have influenced decision-making and the delivery of CPAP in newborn care in the Kenyan health system.

Another limitation of this study is the diversity of healthcare workers present in our focus group discussions, which may have led to reduced reporting of the experiences of the junior members of the healthcare team. Holding such heterogeneous focus group discussions might have meant that the more junior colleagues felt less able to express their opinions due to the presence of their more senior colleagues in a hierarchical healthcare system. Ideally more homogeneous focus groups with regard to seniority or level of education or experience would have been better for more enhanced interactions.[52] However, it was not possible because holding a focus group discussion with junior staff and then a separate one with senior staff due to staff shortages during the period of data collection and would have kept essential staff away from clinical duties for longer.

Finally, we did not include the perspectives of the parents/cargeivers or their families, a key group of stakeholders when evaluating an intervention within a healthcare system. Their perspectives would have given us a better understanding of the acceptability of CPAP in newborn care in Kenya and the effect of partnership between caregivers and healthcare workers on uptake of CPAP.

Nevertheless, to our knowledge, this is the first study in Kenya that has explored in-depth the experiences of healthcare workers using CPAP in newborn care. Our study involved healthcare workers in both the public and private/mission sectors of Kenya, enabling us to identify a diverse range of perspectives from healthcare professionals within the same country but with access to different levels of resources for providing newborn care. These findings will be pertinent for the future scale-up strategies for CPAP and other newborn care technologies in Kenya and other similar settings.

## CONCLUSION

The scale-up of CPAP use in newborn care in Kenya is a strategy that healthcare workers value as a key intervention for improving the survival of newborns. However, there are important challenges that need to be addressed to ensure sustainability and safety of its use in Kenyan hospitals. Worryingly, newborns are currently exposed to preventable risks, limiting the benefit of CPAP in these vulnerable babies. Future CPAP implementation strategies should ensure that infrastructure for CPAP use is fit for purpose, shortages of skilled staff are addressed and that staff training and partnership with caregivers are prioritised.

**Author affiliations**
[1]International Public Health, Liverpool School of Tropical Medicine, Liverpool, UK
[2]Paediatrics Department, Betsi Cadwaladr CHC, Wrexham, UK
[3]International Public Health, Liverpool School of Tropical Medicine, Nairobi, Kenya
[4]General Paediatrics, Gertrude's Garden Children's Hospital, Nairobi, Kenya
[5]Child Health, AMPATH Kenya, Eldoret, Kenya
[6]Neonatal, Child and Adolescent Health, Kenya Ministry of Health, Nairobi, Kenya
[7]Clinical Research, London School of Hygiene and Tropical Medicine, London, UK
[8]Paediatrics and Child Health, University of Nairobi School of Medicine, Nairobi, Kenya
[9]Maternal and Newborn Health, International Health Consulting Services Ltd, Liverpool, UK

**Acknowledgements** We would like to thank the healthcare workers at the Newborn Units of the following hospitals: Aga Khan University Hospital, Nairobi; Bungoma County Referral Hospital; Busia County Referral Hospital; Coast General Hospital, Mombasa; Gertrude's Children's Hospital, Nairobi; Homabay County Referral Hospital; Jaramogi Oginga Odinga Teaching and Referral Hospital, Kisumu; Kakamega County Referral Hospital; Kenyatta National Hospital; Kiambu County Hospital; Kijabe Mission Hospital; Kisii County Referral Hospital; Kisumu County Referral Hospital; Machakos County Referral Hospital; Mama Lucy Hospital, Nairobi; Migori County Referral Hospital; Naivasha Sub-County Hospital; Pumwani Maternity Hospital, Nairobi; Siaya County Referral Hospital; Tenwek Mission Hospital, Bomet. We are grateful also to country health representatives and the Ministry of Health in Kenya for their participation and support in conducting this study. Many thanks to our colleagues Ms Joyce Mutuku, Mr Onesimus Muchemi, Ms Sylvia Gichuru, Ms Linda Omolo, Ms Beatrice Ochieng who supported us with data collection. We thank Dr Sarah White, Ms Sophie Ngugi and Mr Allan Govoga for their important contributions to the conceptualisation of this study. We thank Dr Bernard Olayo and

Miss Caroline Kendi (Centre for Public Health and Development, Kenya) for their advice and support.

**Contributors** HMN coordinated the study, analysed the data, prepared the manuscript and approved the final manuscript as submitted. She took over as Principal Investigator of the study prior to the commencement of data collection and coordinated all aspects of the project thereafter. JLW and MP analysed the data and prepared the manuscript and approved the final manuscript as submitted. FD led the data management of this study, reviewed and approved the final manuscript as submitted. She was the UK-based Research Assistant in this study. PG conceptualised the study, reviewed and approved the final manuscript as submitted. She coordinated the ethics approval process in Kenya. JM conceptualised the study, provided support with the coordination of study activities, reviewed and approved the final manuscript as submitted. She led the communication strategies with the Kenyan stakeholders. MKS and BCN collected, transcribed and analysed the data. They were Research Assistants in this study in Kenya. OHW conceptualised the study, reviewed and approved the final manuscript as submitted. He facilitated the approval of study activities at the national and facility level. JED conceptualised the study, reviewed and approved the final manuscript as submitted. He was the initial Principal Investigator of the study and led the ethics application process. FM conceptualised the study, reviewed and approved the final manuscript as submitted. She was co-Principal Investigator and supported the ethics application processes of the study in Kenya. She also provided very valuable context-relevant information during the data collection and analysis phases. AM supported with study coordination, reviewed and approved the final manuscript as submitted. HS provided guidance on the qualitative methodology during the conceptualisation of the study, reviewed and approved the final manuscript as submitted. She was the lead Social Scientist for this study and provided critical review of the data collection and analysis. MM conceptualised the study, reviewed and approved the final manuscript as submitted. He provided oversight for all the study processes.

**Funding** This project was funded by the UK Department for International Development (Project number: 202549), awarded to Professor Nynke van den Broek.

**Patient and public involvement** Patients and/or the public were not involved in the design, or conduct, or reporting, or dissemination plans of this research.

**Patient consent for publication** Not required.

**Ethics approval** The study was approved by the Research and Ethics Committee at the Liverpool School of Tropical Medicine (protocol number: 15-032) and the University of Nairobi/Kenyatta National Hospital Ethics Research Committee (P56/02/2017).

**Provenance and peer review** Not commissioned; externally peer reviewed.

**Data availability statement** All data relevant to the study are included in the article. Extra data is available by emailing helen.nabwera@lstmed.ac.uk.

**ORCID iDs**
Helen M Nabwera http://orcid.org/0000-0003-1056-729X
Juan Emmanuel Dewez http://orcid.org/0000-0002-5677-8968
Matthews Mathai http://orcid.org/0000-0002-7352-9330

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
