## [Reviewer comments · BMJ Open]

ARTICLE DETAILS

TITLE (PROVISIONAL)	“Sometimes you are forced to play God...”: a qualitative study of healthcare worker experiences of using continuous positive airway pressure in newborn care in Kenya.
AUTHORS	Nabwera, Helen; Wright, Jemma; Patil, Manasi; Dickinson, Fiona; Godia, Pamela; Maua, Judith; Sammy, Mercy; Naimoi, Bridget; Warfa, Osman; Dewez, Juan; Murila, Florence; Manu, Alexander; Smith, Helen; Mathai, Matthews

VERSION 1 – REVIEW

REVIEWER	Tina Slusher University of Minnesota and Hennepin Healthcare USA
REVIEW RETURNED	05-Nov-2019

GENERAL COMMENTS	This is an important paper pointing out the clear need for training among providers with ongoing re-enforcement in the training; the need for lower patient to nurse ratios, and the concerns of both the providers and parents. I would have liked to have had more of a description of the homemade devices in use and am somewhat surprised that more institutions are not using the homemade circuits as outlined in Pocket book of hospital care for children Guidelines for the management of common illnesses with limited resources especially when they ran out of their commercial circuits. It would have been interesting to describe the views of these homemade circuits by providers and parents in places using them especially in those places using both types.
---

REVIEWER	Daniele De Luca South Paris Saclay University - France
REVIEW RETURNED	02-Dec-2019

GENERAL COMMENTS	This is an interesting paper and it might be useful to facilitate CPAP diffusion in the local setting. While this is the clear background and I understand the authors' point, I am not convinced that this would be the right methodology to use. Purely qualitative studies are almost never used in critical care and CPAP represents the first step of modern neonatal critical care. In order to be more "pragmatic" and practically useful I would have expected to see, beside the qualitative part, a more quantitative one with at least some descriptive statistics on the local situation in order to identify critical points and propose solutions. Without this the study remains on the psychological level and may be more suitable for other types of journals (psychology, nursing..) rather than for a high impact general medical journal.
---

	Moreover, relevant guidelines for surveys should also be followed (STROBE or also Burns KE, Duffett M, Kho ME, et al. ACCADEMY Group: A guide for the design and conduct of self-administered surveys of clinicians. CMAJ 2008). On a minor point, it is not correct to state (intro) that CPAP increases the PNX rate. This has been long-time suspected but finally evidence for this is lacking especially in premies. Thus, European(Neonatology 2019) and American (Pediatrics 2014) guidelines do not report this.
--	--

REVIEWER	Andrew Kiragu Children's of Minnesota and Hennepin Healthcare
REVIEW RETURNED	10-Dec-2019

GENERAL COMMENTS	An important study to investigate CPAP use in a representative number of newborn units in Kenya. The study offers insights into the challenges and benefits of the use of CPAP in this LMIC setting. There are a few minor changes that I feel the authors should make. Firstly, they included private and mission hospitals in their study but have not made any clear distinctions between opinions about the use of CPAP in the private and mission hospitals vs. public hospitals and whether or not the same issues faced in public hospitals(staffing, training, appropriate use etc.) are also present in the private and mission setting. In addition, were there any differences between physicians and nurses? Given the significant discrepancy between demand and availability of CPAP, how do the staff manage the rationing that by necessity occurs? Are there any support systems in place for the moral distress faced by these healthcare providers? Overall, a well-written study that will add to the body of knowledge.
--

REVIEWER	Sophie Sarre Florence Nightingale Faculty of Nursing, Midwifery & Palliative Care, King's College London, UK
REVIEW RETURNED	18-Feb-2020

GENERAL COMMENTS	Thank you for giving me the opportunity to review this paper. The case for the importance of this paper is well made. I have only a few relatively minor comments. Limitations Most important is that despite the thoroughness of the methodology and the importance of the findings there are, in my opinion, two important limitations that should be acknowledged: 1) The limitations arising from not including the perspectives of patients and families in the study design. There is a priori a strong justification for their inclusion; but given the findings on their influence on CPAP there is also strong a posteriori justification. 2) The limitations of including such diversity of job role (and in particular, job hierarchy) in the focus groups. Focus groups work best when they are relatively homogenous, as this helps to create a feeling of a safe space. What are the implications of including i.e. nursing students and senior paediatricians in the same focus group, as happened here? Ethics
--

	I would invite some further reflection on the ethical implications of the study. The two limitations noted above suggest some, but there may well be more. Structure of paper The information under the headings Study population and Setting is a bit muddled. The setting (public referral, private ad mission hospitals in Kenya providing CPAP) should be stated clearly before you explain how these hospitals were identified. Page 6 line 53 to p.7 line 31 merely describes findings from a survey. This is not relevant other than as a springboard for exploring the sampling of hospital sites. Similarly, since the survey is not an integral part of this paper Table 2 should not be included. The Sampling section should start with a description of the sampling of the hospitals (which currently appears after the sampling of participants), and it would be useful to include the numbers in the sample and the sampling frame (19/23). I suggest that the paragraph on p13 lines 20-36 currently under 'Shortage of skilled staff...' sits better in the following section, 'Inadequate knowledge and training ...' The relevance of study reference 19 to the Discussion is not clear (p.19 lines 8-13). Minor editorial suggestions I would encourage the authors to limit their use of acronyms as far as possible. There are a number of acronyms that seem necessary / sensible. But adding to them others such as FG (focus group) KII (key informant interview) made it more difficult to read. P4 line 13 it might be worth underlining the relevance of statistics on neonatal deaths in sub-Saharan Africa by pointing out that this includes Kenya, the site of this study. The following phrases are not clear to me "However, these studies were all small scale, supported by external funding. In the absence of such external funding, our data suggests that ..." (Page 19 lines 21-25). What is the relevance of funding? The current study was externally funded. The discussion would benefit from including the implications of the lack of CPAP infrastructure and supplies (rationing, curtailing, re-use) – situations that are vividly drawn in the findings section. Table 1 – Add 'Kenyan' to the table heading I think attention to these comments and suggestions would strengthen what is overall a good paper, which is highly relevant to the provision and delivery of CPAP in Kenya. And with implications for other sub-Saharan and low- and middle-income countries.
--	---

VERSION 1 – AUTHOR RESPONSE

Reviewer: 1

Reviewer Name: Tina Slusher

Institution and Country:

University of Minnesota and Hennepin Healthcare

USA

Please state any competing interests or state 'None declared': None declared

Please leave your comments for the authors below

This is an important paper pointing out the clear need for training among providers with ongoing re-enforcement in the training; the need for lower patient to nurse ratios, and the concerns of both the providers and parents. I would have liked to have had more of a description of the homemade devices in use and am somewhat surprised that more institutions are not using the homemade circuits as outlined in Pocket book of hospital care for children Guidelines for the management of common illnesses with limited resources especially when they ran out of their commercial circuits. It would have been interesting to describe the views of these homemade circuits by providers and parents in places using them especially in those places using both types.

Thank you. The typical homemade devices in use in the two mission hospitals and one county referral hospital were described by healthcare workers as per the recommendations in the WHO Oxygen Therapy for Children manual [1]. We had very limited data on the healthcare worker experiences of using homemade CPAP devices and are therefore not able to report this because we did not set out to explore this. However, we agree that it would be interesting and beneficial to the scale up of CPAP in resource-limited settings to explore the use of these homemade devices in greater depth.

One the limitations of our work (included on page 24, lines 4-9) is that we did not collect data on the experiences of the parents.

We plan to provide detailed descriptions of "homemade" CPAP devices in our related manuscript that will contain the analysis of the quantitative data from this body of work, including the practical details of the different types of CPAP devices in use.

Reviewer: 2

Reviewer Name: Daniele De Luca

Institution and Country: South Paris Saclay University - France

Please state any competing interests or state 'None declared': None declared

Please leave your comments for the authors below

This is an interesting paper and it might be useful to facilitate CPAP diffusion in the local setting. While this is the clear background and I understand the authors' point, I am not convinced that this would be the right methodology to use. Purely qualitative studies are almost never used in critical care and CPAP represents the first step of modern neonatal critical care. In order to be more "pragmatic" and practically useful I would have expected to see, beside the qualitative part, a more quantitative one with at least some descriptive statistics on the local situation in order to identify critical points and propose solutions. Without this the study remains on the psychological level and may be more suitable for other types of journals (psychology, nursing..) rather than for a high impact general medical journal.

Moreover, relevant guidelines for surveys should also be followed (STROBE or also Burns KE, Duffett M, Kho ME, et al. ACCADEMY Group: A guide for the design and conduct of self-administered surveys of clinicians. CMAJ 2008).

Thank you. The data we present here is the qualitative component of a mixed methods study. We have added these details to the methods section (page 6, lines 1-2).

The implementation of an intervention such as CPAP in a low resource setting is challenging because it requires a minimum level of resource allocation (including staffing and physical infrastructure), which is often not available. We agree that quantitative data with quantitative analyses are very important, but in the case of complex interventions, such studies need to be complemented by

qualitative studies. Qualitative studies aim to explore in greater depth complex aspects of implementation, including the views of the frontline workers in order to understand the barriers and facilitators to support quality improvement and future scale up strategies. Our wider project includes both quantitative and qualitative data. Due to the large volume of data, it was not possible to incorporate both types of data into one manuscript. We therefore opted to disseminate these important data as separate qualitative (this manuscript) and quantitative (in preparation) manuscripts, which we are preparing in line with the STROBE guidelines for cross-sectional surveys. This paper is about the in-depth qualitative data of this project. For this qualitative paper we followed the Consolidated criteria for reporting qualitative research (COREQ) that are validated guidelines for reporting qualitative studies. We felt that highlighting the healthcare worker experiences was an important first step in describing the operational aspects of this key intervention in hospitalised newborns in a resource constrained country - Kenya, which is keen to scale up this intervention. However, to provide some contextual information, we included some of the quantitative data in this manuscript, which we have now excluded and will reserve for the quantitative manuscript.

An example of how qualitative and quantitative data can be used to address research questions on complex interventions in health systems in a low and middle income country setting can be found in the following papers, which assessed the use of CPAP in neonates in India (the first is a qualitative study <https://www.ncbi.nlm.nih.gov/pmc/articles/PMC6220518/> that was formative and complemented the quantitative study <https://bmjopen.bmj.com/content/10/2/e031128> as part of the same mixed methods project). These data are now being used to inform the scale up strategies of CPAP in India. We anticipate that our data will also enable the Ministry of Health in Kenya and other partners in the implementation of CPAP in Kenya to use these data to support future scale up of this vital intervention in newborn care.

On a minor point, it is not correct to state (intro) that CPAP increases the PNX rate. This has been long-time suspected but finally evidence for this is lacking especially in premies. Thus, European (Neonatology 2019) and American (Pediatrics 2014) guidelines do not report this.

Thank you. We have deleted "pneumothorax" from the list of potential adverse events associated with using CPAP in newborn care. (page 5, line 2)

Reviewer: 3

Reviewer Name: Andrew Kiragu

Institution and Country: Children's of Minnesota and Hennepin Healthcare

Please state any competing interests or state 'None declared': None declared

Please leave your comments for the authors below

An important study to investigate CPAP use in a representative number of newborn units in Kenya. The study offers insights into the challenges and benefits of the use of CPAP in this LMIC setting. There are a few minor changes that I feel the authors should make. Firstly, they included private and mission hospitals in their study but have not made any clear distinctions between opinions about the use of CPAP in the private and mission hospitals vs. public hospitals and whether or not the same issues faced in public hospitals (staffing, training, appropriate use etc.) are also present in the private and mission setting. In addition, were there any differences between physicians and nurses? Thank you. From our data it was difficult to draw out clear distinctions of healthcare worker experiences of CPAP use in private and mission vs public hospital due to the limited number of private and mission hospitals that we were able to access for this study. However, with the data that we have, we have highlighted the key differences in the healthcare worker experiences in page 10, lines 7-9 and page 15, lines 1-7. The challenges with implementation of CPAP were predominantly experienced in the public sector. To help distinguish the experiences, we have now provided details of

whether the quotes were from healthcare workers in the private/mission hospitals vs public hospitals. Similarly, as this data was collected during a period of multiple healthcare worker strikes, it was very difficult to have separate focus group discussions for physicians vs nurses. In addition, often the physicians were not available for the key informant interviews and therefore nominated a senior nurse to be interviewed in their place. It was therefore not possible to draw out key differences in experiences of CPAP use between nurses and physicians as the data was predominantly from the nurses working at the hospitals. We have highlighted these issues under the section on the limitations of this study (pages 23 & 24).

Given the significant discrepancy between demand and availability of CPAP, how do the staff manage the rationing that by necessity occurs? Are there any support systems in place for the moral distress faced by these healthcare providers?

Thank you. We have reported how staff managed the rationing under on Page 10, lines 19-21, page 11, lines 1-12: where staff explain how they sometimes have to remove newborns from CPAP machines when they improved even if they have not met the criteria for stopping CPAP in order to initiate it on a sicker newborn. Although the staff did not mention any support systems in place for staff facing the moral distress of CPAP rationing in newborn care, one hospital mentioned the presence of "CPAP champions". Their role included empowering their fellow nurses through training and providing practical support/advice to help build their confidence of using CPAP in newborn care. This is highlighted on page 16, lines 11-18.

Overall, a well-written study that will add to the body of knowledge.

Reviewer: 4

Reviewer Name: Sophie Sarre

Institution and Country: Florence Nightingale Faculty of Nursing, Midwifery & Palliative Care, King's College London, UK

Please state any competing interests or state 'None declared': None declared

Please leave your comments for the authors below

Thank you for giving me the opportunity to review this paper. The case for the importance of this paper is well made. I have only a few relatively minor comments.

Limitations

Most important is that despite the thoroughness of the methodology and the importance of the findings there are, in my opinion, two important limitations that should be acknowledged:

1) The limitations arising from not including the perspectives of patients and families in the study design. There is a priori a strong justification for their inclusion; but given the findings on their influence on CPAP there is also strong a posteriori justification.

Thank you. We appreciate that the experience of parents/caregivers and families is pertinent in a project of this nature. For this phase of the project the focus was on healthcare worker perspectives on the use CPAP in newborn care. This is an important limitation and have included this in the relevant section on page 24, lines 5-9.

2) The limitations of including such diversity of job role (and in particular, job hierarchy) in the focus groups. Focus groups work best when they are relatively homogenous, as this helps to create a feeling of a safe space. What are the implications of including i.e. nursing students and senior paediatricians in the same focus group, as happened here?

Thank you. We agree this is another very important limitation that we have outlined in the relevant section on page 23, lines 15-21, page 24, lines 1-4.

Ethics

I would invite some further reflection on the ethical implications of the study. The two limitations noted above suggest some, but there may well be more.

Thank you. We conducted this work in a year when there were multiple healthcare worker strikes therefore many newborn baby units were already short staffed and thus our study could have had a negative impact by disrupting further clinical activities. However, we worked closely with the Ministry of Health, county governments and hospitals to ensure that we only collected our data on days when the units reported having adequate numbers of staff and could function for 1-2 hours with fewer staff to enable other colleagues attend our interviews. On many occasions we cancelled planned visits and waited for a more appropriate time. We have added these details under the methods section (page 7, lines 20-21 & page 8, lines 1-5).

During the process of data collection, there were instances where some healthcare worker perspectives highlighted the gaps in knowledge regarding the use of CPAP. A requirement of the ethics approval process was that we anonymised the responses so that individual healthcare workers were not identified and we therefore not able to inform individual heads of department about these knowledge gaps. However, in the summary report that we produced for the Ministry of Health, we highlighted the need for training followed by refresher training for staff involved in delivering CPAP to hospitalised newborns in Kenya.

Structure of paper

The information under the headings Study population and Setting is a bit muddled. The setting (public referral, private ad mission hospitals in Kenya providing CPAP) should be stated clearly before you explain how these hospitals were identified. Page 6 line 53 to p.7 line 31 merely describes findings from a survey. This is not relevant other than as a springboard for exploring the sampling of hospital sites. Similarly, since the survey is not an integral part of this paper Table 2 should not be included. Thank you. We have excluded all the data from the survey that we will incorporate into a separate manuscript that focuses on the survey findings.

The Sampling section should start with a description of the sampling of the hospitals (which currently appears after the sampling of participants), and it would be useful to include the numbers in the sample and the sampling frame (19/23).

Thank you. We have revised this in page 7, lines 3-5 and provided the details of how the hospitals were sampled.

“There were two phases of sampling in this study. Convenience sampling was used to recruit the 19 hospitals, which were the hospitals that gave us permission to access their facilities and staff to conduct our study.”

I suggest that the paragraph on p13 lines 20-36 currently under ‘Shortage of skilled staff...’ sits better in the following section, ‘Inadequate knowledge and training ...’

Thank you. We have moved the section, ‘Inadequate knowledge and training ...’(page 15, lines-21, page 16, lines1-5)

The relevance of study reference 19 to the Discussion is not clear (p.19 lines 8-13).

We have retained the study reference 19 as it highlights the fact that in when CPAP was used in newborn care in a Kenyan mission hospital, where the relevant structures were in place to support its use, there was improved survival among their preterm infants. We have highlighted this point now (page 19, lines17-20).

Minor editorial suggestions

I would encourage the authors to limit their use of acronyms as far as possible. There are a number of acronyms that seem necessary / sensible. But adding to them others such as FG (focus group) KII (key informant interview) made it more difficult to read.

Thank you. We have removed the less commonly used acronyms from our manuscript to improve readability.

P4 line 13 it might be worth underlining the relevance of statistics on neonatal deaths in sub-Saharan Africa by pointing out that this includes Kenya, the site of this study.

We have emphasised this on page 4, line 6 with more details on page 5, lines 4-14 to highlight the relevance of choosing Kenya as the site for conducting this work

“The majority of these neonatal deaths occurred in low and middle-income countries (LMICs), with the highest burden in South Asia and sub-Saharan Africa (including Kenya)”

The following phrases are not clear to me “However, these studies were all small scale, supported by external funding. In the absence of such external funding, our data suggests that ...” (Page 19 lines 21-25). What is the relevance of funding? The current study was externally funded.

Here we meant that the implementation of CPAP in newborn care in these studies was supported by external funding that provided the relevant equipment, staff support, training and monitoring. When these projects came to an end, the systems that were set up to ensure that appropriate standards were maintained often were not sustained. We have revised this statement to clarify this point. (page 20, lines 2-6)

The discussion would benefit from including the implications of the lack of CPAP infrastructure and supplies (rationing, curtailing, re-use) – situations that are vividly drawn in the findings section.

We agree and have expanded on these details in the discussion now. (page 20, lines 7-21)

Table 1 – Add ‘Kenyan’ to the table heading

Thank you. We have added this. (page 33)

VERSION 2 – REVIEW

REVIEWER	Andrew Kiragu Children's Minnesota and Hennepin Healthcare, USA
REVIEW RETURNED	20-Apr-2020

GENERAL COMMENTS	As noted previously, an important addition to the literature regarding CPAP use in low and middle-income countries. I am satisfied that the authors have provided clarifying responses and made adequate revisions to the manuscript to move forward with acceptance for publication.
---

REVIEWER	Sophie Sarre King's College London, UK
REVIEW RETURNED	19-Mar-2020

GENERAL COMMENTS	I am happy with the changes made and recommend publication.
---